# Fire Protection of Steel Structures with Epoxy Coatings under Cryogenic Exposure

**Marina Gravit** [1] , **Boris Klementev** [2] **and Daria Shabunina** [1,*]

1   Civil Engineering Institute, Peter the Great St. Petersburg Polytechnic University, 195251 St. Petersburg, Russia; marina.gravit@mail.ru
2   Department of Expertise, LLC "Arctic LNG 2", 117393 Moscow, Russia; zeugmas@icloud.com
*   Correspondence: d.shabunina00@gmail.com

**Abstract:** Cases of fire with highly flammable, combustible liquids and combustible gases with high potential heat emission at oil and gas facilities are assumed to develop as a hydrocarbon fire, which is characterized by the temperature rising rapidly up to 1093 ± 56 °C within five minutes from the test start and staying within the same range throughout the test, as well as by overpressure being generated. Although various fireproof coating systems are commonly used to protect steel structures from high temperatures, a combination of fire protection and cryogenic spillage protection, i.e., protection from liquefied natural gas (LNG), is rather an international practice novelty regulated by standards ISO 20088. Thanks to their outstanding features, i.e., ability to sustain chemical and climatic impact, these epoxy-based materials are able to ensure positive fireproof performance for steel structures in the case of potential cryogenic impact. The article discusses tests on steel structures coated with epoxy fireproof compounds, specifically PREGRAD-EP, OGRAX-SKE and Chartek 2218. The test records show the time from the start of cryogenic exposure to the said sample reaching the limit state, as well as the time from the start of heat impact to the sample reaching the limit state in case of hydrocarbon fire temperature.

**Keywords:** steel structure; oil and gas facilities; liquefied natural gas (LNG); cryogenic spillage protection; passive fire protection (PFP); epoxy syntactic materials; hydrocarbon fire

## 1. Introduction

The oil and gas industry is one of the global economy's leading and most challenging branches, and is of critical importance [1]. In most cases, energy economy assets and facilities (buildings, structures and equipment) are deemed higher hazard sources as such process facilities imply handling and storing considerable volumes of combustible and explosive substances, and non-compliance with the relevant safety rules may entail fire, explosion and/or spillage [2–6]. The major and most severe incidents at Piper Alpha, the North Sea, (6 July 1988) and Deepwater Horizon (20 April 2010) proved that the standards adopted in the oil production industry should be improved [7,8].

The following temperature cases are stipulated to be used for standardized structural fire tests: "standard" (cellulosic) fire; external, slow heating; and hydrocarbon fire [9]. Steel structures in a fire or blast emergency scenario in oil and gas facilities suffer high-temperature and overpressure impact corresponding to the hydrocarbon fire case. During the several minutes when the fire starts, the temperature reaches 1000 °C and higher [10–12]. The steel structure strength becomes drastically lower within the range of 400–600 °C, while in case of applied load, the unprotected structure almost immediately loses its stability [13]. For this reason, structures that can withstand, e.g., higher temperatures and blast shockwaves, are protected with fireproofing, must be used at hazardous facilities.

Standards such as EN 1473 and NFPA 59A [14,15] or industrial standards of the major oil and gas companies stipulate that the steel structures being part of equipment, process units and piperacks related to processing/storing liquefied natural gas (LNG), shall be

resistant to cryogenic exposure, i.e., to the impact of gas liquefied by compression, staying at ultralow (cryogenic) temperatures below −150 °C [16–18], affecting to the destructive impact of the integrity of steel parts with embrittlement temperatures ranging from −20 °C to −40 °C, and structural connections, such as welded joints [19,20]. An LNG spill can cause a high-pressure heat load fire. There are two common types of fire that can happen in hydrocarbon processing facilities: pool fire, which occurs when a flammable liquid leaks from a vessel or pipeline, forming a "pool" of liquid that ignites, and jet fire, which is a potentially more dangerous type of fire that can result from the rupture of a pressurized vessel and/or pipeline (Figure 1) [21].

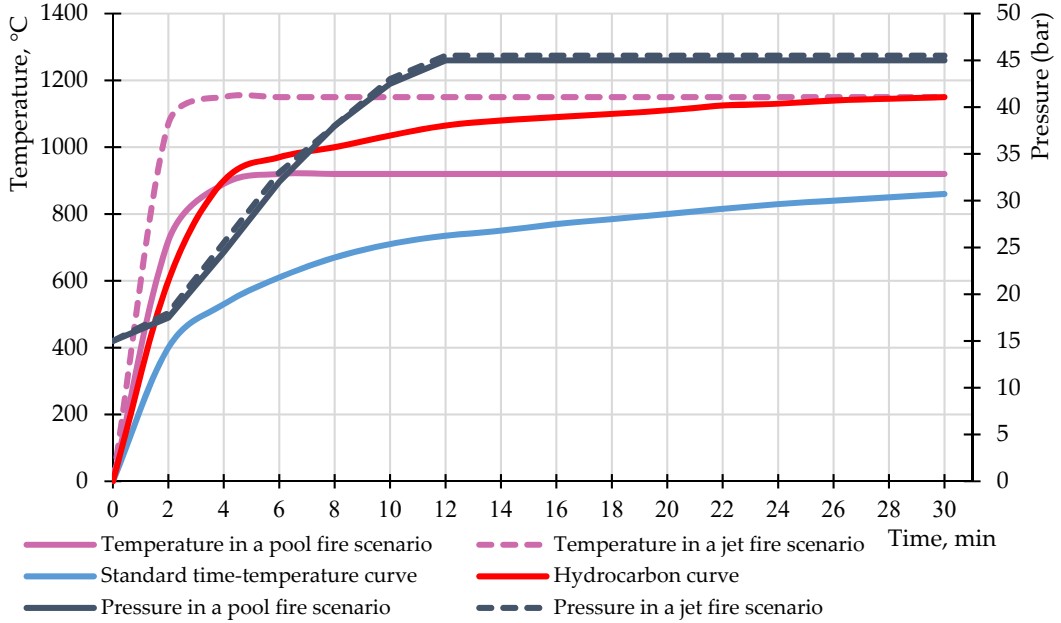

**Figure 1.** Results for 0.5 m diameter vessel with 45 bar design pressure of two types of fire: pool fire and jet fire [21].

The risks of exposure to cryogenic liquids must be considered when designing steel structures for oil and gas facilities. For example, [22] shows the cryogenic risk analytics and suggests a method that improves the quantitative evaluation of the cryogenic risk to determine measures to reduce and optimize the cryogenic spillage protection coating in cryogenic-hazardous areas of an oil and gas industrial unit. The authors of [23] offer a simplified method of defining the potential cryogenic spillage area at an LNG production plant, confirmed by experimental tests and simulations conducted to determine the degree of the danger zone depending on the size of the leakage hole.

Fireproof materials that maintain a steel structure's integrity and heat insulation within the temperature range of −200 °C to 1300 °C need to be used for protecting the steel structure's oil and gas assets, especially those producing LNG, not only from fire but also from cryogenic spills [24].

One of the methods to prevent fire spread and ensure the stability of buildings and structures in case of fire is applying passive fire protection (PFP), which encompasses dedicated fireproof plasters, paints, casings, slabs and intumescent paints [25]. At O&G production sites, intumescent coatings based on epoxy binding agents are of wide and common use, with chemical and climatic resistivity, a low volatile substance content, a life cycle of at least 25 years, superb adhesion and high repairability as their key characteristics [12,26]. Intumescent coatings swell when exposed to high temperatures to create foamcoke. The coating grows to form a thermal barrier. By increasing in volume and decreasing in density, intumescent coatings slow down the heating of steel and increase the time needed to destroy steel structures [27]. Epoxy products, used as the PFP of steel structures and equipment for many years, have proved their durability and reliability in

hydrocarbon fire mode in marine conditions. Intumescent coatings are an effective PFP for steel structures in high-risk situations in petrochemical plants and offshore platforms [28]. At the same time, [29] proposes an experiment to evaluate the charring strength of silicone and epoxy-based intumescent coatings applied to steel panels in a hydrocarbon fire test.

Insulating materials based on epoxy binder can be used as protection against cryogenic spills, compatible with PFP. For example, [30] experimentally proved that PFP epoxy products could effectively solve the problem of cryogenic filling with or without hydrocarbon ignition; in addition, weather resistance tests showed that epoxy syntactic materials could provide excellent protection against corrosion of the main steel structure. The shock load jump of LNG tankers is an important problem that can cause damage to the LNG cargo containment system, which results in cryogenic leakage from the ship's hull. An insulation system based on epoxy resin is an acceptable material that can withstand repetitive shock loads [31]. In [32], an efficient method for obtaining epoxy nanocomposites with excellent mechanical properties at cryogenic temperatures is proposed, significantly increasing the cryogenic tensile strength and impact toughness of epoxy nanocomposites.

Being certified by a major oil and gas company is one of the guarantees of high quality for a fireproofing material. For example, the Gazprom register includes OGRAX-SKE epoxy coating (Unichimtek, Russia), the Rosneft register includes PREGRAD-EP epoxy coating (PREGRAD, Russia) [33] and Chartek 2218 (AkzoNobel, Netherlands) epoxy coating is recommended to be used at O&G assets and has been adopted by NOVATEK PJSC.

This article considers the behavior of flame-retardant coatings used on the structures of various profiles of oil and gas facilities as PFP, under different scenarios of cryogenic liquid spill (PREGRAD-EP and OGRAX-SKE when samples are completely immersed in liquid nitrogen, Chartek 2218 when the sample is exposed to two-phase cryogenic exposure) with subsequent hydrocarbon fire tests.

## 2. Materials and Methods

The test methods for liquid hydrocarbon cryogenic emissions of various natures are specified in the ISO 20088 series [34,35], where liquid nitrogen is used as an equivalent for liquid hydrocarbons due to having lower boiling temperature than that of liquefied natural gas or liquid oxygen, and due to not being flammable.

The ISO 20088-1:2016 [34] implies submerging the test sample fully into the cryogenic liquid. The limit temperature decrease is defined as the difference between the ambient temperature and the limit temperature for the steel. For example, the limit temperature of structural steel used for ship hulls or for equipment is $-40\ ^\circ$C. The sample complies with the requirements subject to not exceeding the limit temperature. ISO 20088-3:2018 [35] outlines the method of defining the protection system's resistivity to a cryogenic spray jet stemming from a pressurized emission that does not generate conditions equivalent to immersion. A cryogenic jet can be caused by a discharge from pressurized process equipment. High pressure leads to a high impulse that, together with the extreme cryogenic temperature, can potentially jeopardize the cryogenic spill protection. The test described herein is characteristic for LNG discharge through a hole of 20 mm diameter or less. Both the technical literature analytics data [22,23,30,36] and practical experience show that the two-phase spray is the most frequent cryogenic impact case.

The tests aimed at defining the time required to reach the critical state under cryogenic and subsequent fire impact with two fire protection coatings (PREGRAD-EP, sample no. 1.1 and sample no. 1.2; OGRAX-SKE, sample no. 2). They were held applying the methods of the POZH-AUDIT research center (Russia) based on ISO 20088-1:2016 [34], considering a decreased limit temperature of $-60\ ^\circ$C, according to the client's technical assignment; as for the Chartek 2218 compound (sample no. 3), the tests were held in compliance with ISO 20088-3:2018 [35]. These coatings are corrosion-resistant and can be applied at the construction site by airless spraying. The features and characteristics of the investigated flame retardant coatings are presented in Table 1.

**Table 1.** Features and characteristics of the researched fireproofing coatings.

| Features and Characteristics | Samples No. 1.1 and No. 1.2 | Sample No. 2 | Sample No. 3 |
|---|---|---|---|
| Base | modified epoxy resins and dedicated aggregates | | |
| Color, finish | light grey | from grey to black | light grey |
| Density, kg/L | $0.9 \pm 0.05$ | $1.3 \pm 0.2$ | 1.0 |
| Solids, % | $97 \pm 1$ | $93 \pm 3$ | 100 |
| Application humidity, not higher than, % | 80 | 90 | 85 |
| Application temperature, not lower than, °C | −10 | +5 | +10 |

Before the tests, measurements were made of the actual thicknesses applied to the samples of fire-retardant coating, which were measured at 36 points along the perimeter of the heated surface, in steps of 500 mm in height of the samples, the results of which were taken as the arithmetic mean value.

Points when the metal of sample no. 1.1, sample no. 1.2 and sample no. 2 reached the critical temperature of −60 °C, or −45 °C for sample no. 3, are taken as the relevant limit state points. During the hydrocarbon fire test, the sample metal reaching the critical temperature of 500 °C is taken as the relevant limit state [37].

Upon completion of the cryogenic impact session sample no. 1.1, sample no. 1.2 and sample no. 2 were retrieved from the liquid nitrogen, and the coatings were inspected for cracking, blistering or peeling. Then the samples were placed in the furnace for fire tests, and ultimately hydrocarbon fire heat impact was applied, as follows (1):

$$T - T_0 = 1080 \times \left(1 - 0.325 \times e^{-0.167t} - 0.675 \times e^{-2.5t}\right) \tag{1}$$

where $T$ is the temperature inside the furnace in °C, corresponding to the relevant time t; $T_0$ is the temperature in °C inside the furnace prior to the start of the heat impact; and $t$ is the time in min from the start of the test.

Cryogenic test conditions were carried out according to [34,35], and fire test conditions were following with ISO 834-1:1999 [38].

### 2.1. Experiment No. 1

Two samples were examined for experiment no. 1. Sample no. 1.1 was a steel I-beam column, profile 20B1 [39], with a section ratio of 294 mm$^{-1}$ [40]; its length was 1700 mm and protective layer thickness was 18 mm.

Sample no. 1.2 was a steel I-beam column, profile 20B1 [39], with a section ratio of 294 mm$^{-1}$ [40]; its length was 1700 mm, and protective layer thickness was 27 mm. Reinforcing mesh and glyphthalic primer were used for both samples.

The method implies stepwise definition of the time from the start of cryogenic exposure on the test sample until the limit state is reached in the cryogenic test case, and subsequently, defining the time from the start of heat impact on the test sample to this sample reaching the limit state in the hydrocarbon fire test case.

The samples, each having three thermocouples installed under the fireproof coating by fullering/caulking in the midsection of the I-beam web and on the inner surface of the I-beam flanges per [37], were exposed to shock cooling as they were put into a dedicated vessel which was then filled with liquid nitrogen. The temperature of the steel layer under the coating was registered, and the appearance alterations were visually monitored.

### 2.2. Experiment No. 2

Sample no. 2 was a 100 × 100 × 8 mm steel square tube [41] with a section ratio of 134 mm$^{-1}$ [40], with a dry layer total thickness of 22.5 mm. Glyphthalic primer (dry layer thickness at least 0.05 mm) and reinforcing mesh were used.

The essence of the method in question, which implies consecutive cryogenic and fire tests, was to define the sample temperature (average value derived from the readings obtained at all the thermocouples installed on the sample): 10 min from the start of cryogenic impact in the liquid nitrogen, 120 min from the start of heat impact on the sample in hydrocarbon fire conditions.

The sample was initially exposed to cryogenic impact. The liquid nitrogen level was maintained at an elevation of 800 mm from the vessel bottom, with the sample itself immersed in the nitrogen to a length of 750 mm. The control over the liquid nitrogen level was ensured through a thermocouple installed at an elevation of 800 mm from the bottom of the coolant vessel (Figure 2).

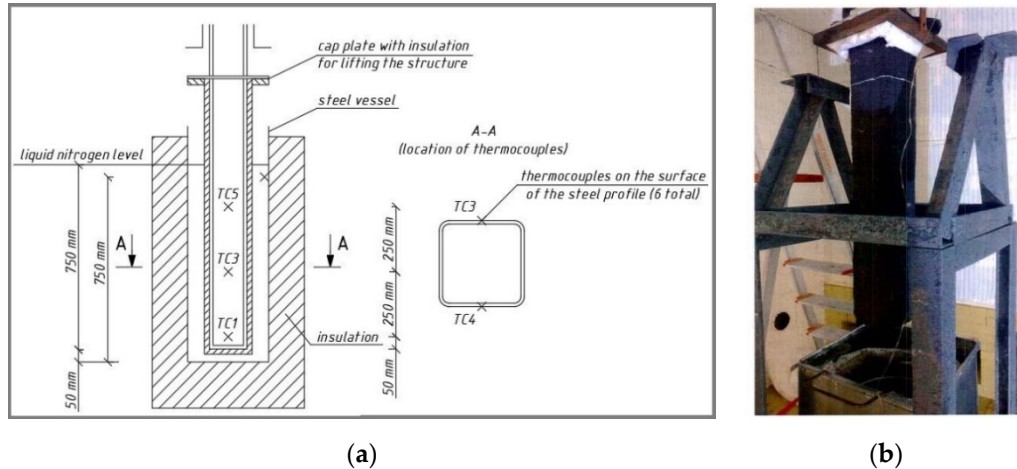

(**a**)　　　　　　　　　　　　　　　　　　　　　　　　　　(**b**)

**Figure 2.** (**a**) Schematic of the test unit for the cryogenic experiment; (**b**) Test unit for the cryogenic experiment.

After completion of cryogenic exposure, the sample was removed from the vessel with liquid nitrogen, checked for coating defects, and then subjected to heat exposure.

### 2.3. Experiment No. 3

Sample no. 3 was a steel I-beam column with a section ratio of 295 mm$^{-1}$ [40], its length was 1700 mm, and protective layer thickness was 8.7 mm. A reinforcing mesh and glyphthalic primer base coat were used. The location of the thermocouples on the sample was similar to experiment no. 1

The method implies defining the time period consecutively from the start of cryogenic impact until a drop in temperature of 50 °C is reached (to −45 °C), and subsequently defining the time period from the start of heat impact on the test sample in a hydrocarbon fire test case, until the sample reaches the limit state.

Sample no. 3 was exposed to shock cooling resulting from two-phase liquid nitrogen impact, as the temperature was recorded using thermoelectric transducers located on the sample in a pattern similar to experiment no. 1; cracking generation and spread were visually monitored (Figure 3).

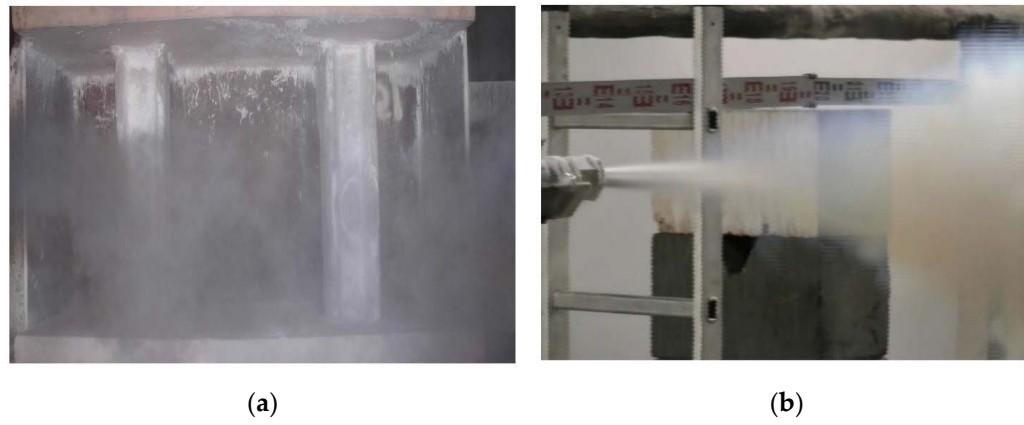

(**a**)                                                                (**b**)

**Figure 3.** (**a**) I sections coated with Chartek after exposure to two-phase spray exposure; (**b**) two-phase spray generating a condition at the steel surface equivalent to immersion in a cryogenic liquid at −145 °C.

## 3. Results

### 3.1. The Results of Experiment No. 1

According to the test results, it was found that sample no. 1.1, with its dry layer thickness of 18 mm, as applied to the 20B1 I-beam column, its length 1700 mm with a section ratio of 294 mm$^{-1}$, ensured a time period of 31 min until the critical temperature of −60 °C was reached on the sample, fully immersed in liquid nitrogen, and ensured fire protection efficiency for 120 min at hydrocarbon fire temperatures.

According to the test results, it was found that sample no. 1.2, with its dry layer thickness of 27 mm, as applied to the 20B1 I-beam column, its length 1700 mm and with a section ratio of 294 mm$^{-1}$, ensured a time period of 67 min until the critical temperature of −60 °C was reached on the sample (Figure 4), fully immersed in liquid nitrogen, and ensured fire protection efficiency for 120 min at hydrocarbon fire temperatures.

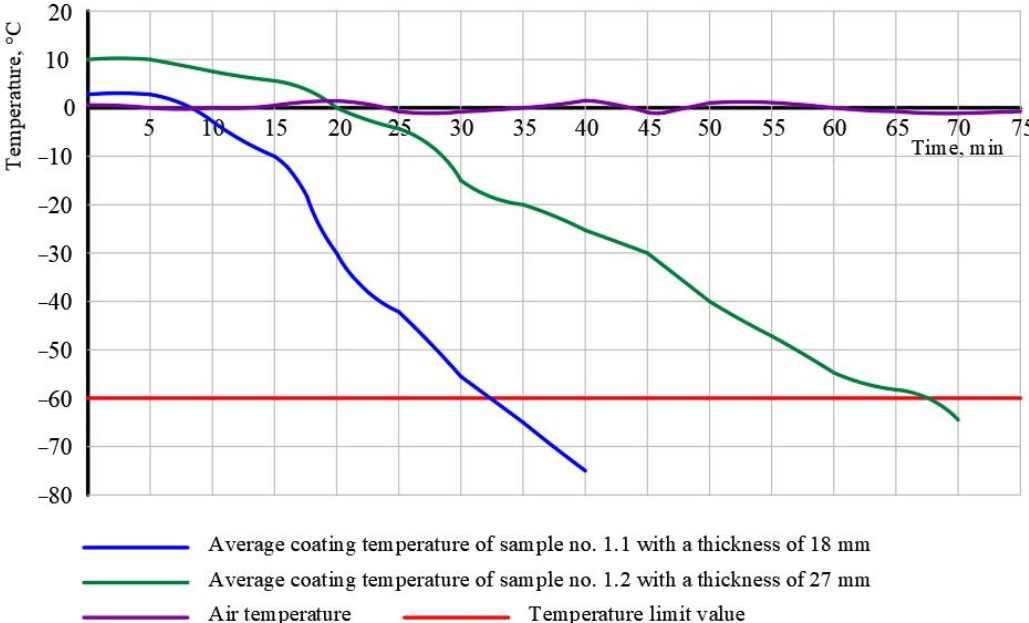

**Figure 4.** Cryogenic test temperature curves for sample no. 1.1 and sample no. 1.2.

The cryogenic exposure on sample no. 1.1 and sample no. 1.2 was stopped after 31 min and 67 min, respectively, upon reaching the critical temperature, while the fire tests

that followed were stopped after 125 min without reaching the critical temperature on the samples.

### 3.2. The Results of Experiment No. 2

According to the test results, it was found that sample no. 2, with its dry layer thickness of 22.5 mm applied on a square cross-section steel pipe (100 × 100 × 8 mm) and with a section ratio of 134 mm$^{-1}$, ensured fire protection performance under hydrocarbon fire for at least 120 min after 10 min of cryogenic impact on the sample fully immersed in liquid nitrogen.

During the cryogenic impact period, the average temperature of the sample was never lower than 40 degrees below its initial temperature. Upon completion of the cryogenic impact, the average temperature of the sample was −18 °C. After the cryogenic test, the fireproof coating had non-through cracks, not more than 0.5 mm wide. No blistering, peeling or other visible defects were found on the fireproof coating after the cryogenic exposure.

During the fire test at 15 min, foamcoke began to form, protecting the structure from heat. As soon as the required time was reached, the test was stopped. The average temperature registered on the sample was 468 °C. After the completion of heat exposure, it was found that the formed foamcoke maintained its structure and integrity.

### 3.3. The Results of Experiment No. 3

According to the test results, it was found that sample no. 3, with its dry layer thickness of 8.7 mm, as applied to the I-beam column, its length 1700 mm and with a section ratio of 295 mm$^{-1}$, ensured fireproof performance under hydrocarbon fire impact for at least 120 min, after a 30-min cryogenic impact period with liquid nitrogen two-phase spray.

During the cryogenic impact period, the average temperatures of the sample dropped by more than 50 degrees against the ambient temperature (down to −45 °C). The temperature difference did not entail any cracking of the fireproof coating.

The fire test that followed the cryogenic test for sample no. 3 was completed after 120 min without reaching the critical temperature.

## 4. Discussion

Analyzing the test methods of steel samples for cryogenic exposure and subsequent fire testing (Table 2), we can see that the methods of Russian entities use the method of full immersion of the sample in a cryogenic liquid while setting a lower limit temperature (−60 °C) according to the customer requirements, while international companies determine the resistance of the fireproof coating when released under pressure from the cryogenic spray until the sample temperature decreases by more than 50 °C relative to the ambient temperature. Additionally, fire tests for hydrocarbon fire mode in international methods are conducted according to UL 1709 [42], and in Russian—according to GOST R EN 1363-2-2014 (EN 1363-2:1999) [9,43], which differ from each other in temperature and time dependence [44]. Experiments carried out according to customer requirements differ in the number of samples, the profiles used (square cross-section or I-beams) and the section ratio. Figure 5 shows the time–temperature curves of the samples during the fire test.

The time–temperature curves presented in Figure 5 show that all epoxy intumes-cent coatings subjected under different scenarios to cryogenic exposure under liquid nitrogen conditions did not reach the critical temperature of 500 °C after 120 min, which proves their effectiveness when used not only as PFP, but also as a combined protection against cryogenic temperatures with subsequent fire exposure with in-creased pressure and heat flow at oil and gas facilities. In the process of fire testing the flame retardant coating of sample no. 2, a smooth increase in temperature was ob-served throughout the test, while in sample no. 1.1 and sample no. 1.2 a sharp increase in temperature was recorded almost from the first minutes of the experiment with subsequent "linear" growth due to different coefficients of the swelling of coatings and thermal conductivity of foamcoke, which is probably due to chemical differences in the coatings, as well as different section ratios.

**Table 2.** Comparison between testing methods of fireproof coatings.

| Comparison Criteria/Materials | Sample No. 1.1/Sample No. 1.2 (Own Method Based on [34]) | Sample No. 2 (Own Method Based on [34]) | Sample No. 3 (per [35]) |
|---|---|---|---|
| Section ratio, mm$^{-1}$ | 294 | 134 | 295 |
| Fireproof coating application | enamel base coat + fireproof coating of required thickness + reinforcing fiberglass mesh | base coat + fireproof coating + reinforcing fiberglass mesh + fireproof coating + reinforcing fiberglass mesh + fireproof coating | base coat + fireproof coating + reinforcing fiberglass mesh + fireproof coating |
| Thickness of dry layer of the fireproof coating, mm | (a) 18.0 (b) 27.0 | 22.5 | 8.7 |
| Cryogenic impact test | | | |
| (a) Medium | liquid nitrogen | liquid nitrogen | liquid nitrogen |
| (b) Method of exposing the sample to liquid nitrogen | full immersion | full immersion | two-phase impact |
| (c) Test method | test held within 31 and 67 min until the critical temperature of −60 °C was reached | test held within 10 min (without reaching the critical temperature of −60 °C) | test held within 30 min until the critical temperature of −45 °C was reached |
| Fire impact test | | | |
| (a) Start of fire test | after cryogenic exposure | after cryogenic exposure | after cryogenic exposure |
| (b) Fire type | hydrocarbon fire | hydrocarbon fire | hydrocarbon fire |
| (c) Test method | completed after 125 min without reaching the critical temperature | completed after 120 min without reaching the critical temperature | completed after 120 min without reaching the critical temperature |

Average coating temperature of sample no. 1.1 with a thickness of 18 mm and a section ratio of 294 mm$^{-1}$
Average coating temperature of sample no. 1.2 with a thickness of 27 mm and a section ratio of 294 mm$^{-1}$
Average coating temperature of sample no. 2 with a thickness of 22.5 mm and a section ratio of 134 mm$^{-1}$
Temperature limit value   Standard time-temperature curve   Hydrocarbon curve

**Figure 5.** Fire test temperature curves for samples.

## 5. Conclusions

Research devoted to cryogenic spilling on fireproof coatings of steel structures is extremely sparse at the moment, as the technology of hydrocarbon liquefaction has appeared quite recently, but it wide interest in this study is expected, given the expansion of the world oil and gas complex in the Arctic and Antarctic. Many manufacturers of fireproofing products for steel structures design their own methods of testing where only local cryogenic spills are considered. The differences between the methods of cryogenic impact tests (local spillage, immersion, two-phase spray) and subsequent hydrocarbon fire do not allow the assessment of fire protection means properly or allow the carrying out of benchmark analysis. Due to the high pressure of the cryogenic jet, the impulse may, together with extreme cryogenic temperatures, jeopardize the cryogenic spill protection. Research shows that the most likely cryogenic exposure scenario is the two-phase spray described in ISO 20088-3:2018, but full immersion in a cryogenic liquid of a structure protected by a flame retardant, shown in ISO 20088-1:2016, is a more extreme method and identifies the most effective coatings. In the case of an accident, any scenario of cryogenic effects on building structures is possible. Thus, fire retardant coatings should be tested simultaneously for both full immersion in a cryogenic environment and for two-phase spray. A more detailed study of cryogenic spillage on the fire protection efficiency of flame retardants of different chemical nature is required; thus, the authors will plan further research in this field.

**Author Contributions:** Conceptualization, M.G.; Investigation, B.K.; Data curation, D.S. All authors have read and agreed to the published version of the manuscript.

**Funding:** The research is partially funded by the Ministry of Science and Higher Education of the Russian Federation under the strategic academic leadership program 'Priority 2030' (Agreement 075-15-2021-1333 dated 30.09.2021).

**Institutional Review Board Statement:** Not applicable.

**Informed Consent Statement:** Not applicable.

**Data Availability Statement:** Testing laboratory of the Federal State Budgetary Institution "POZH-AUDIT" and Technical Note for the Cryogaz-Vysotsk LLC Project.

**Conflicts of Interest:** The authors declare no conflict of interest.

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
