# Peer review of "Fire Protection of Steel Structures with Epoxy Coatings under Cryogenic Exposure"

_buildings, doi:10.3390/buildings11110537_

Round 1
Reviewer 1 Report
The author focused on the fire resistance of steel structures with different fire protection methods. The synergism or antagonism of fire protection and cryogenic spillage protection for steel should be analyzed. The cryogenic spillage protection on the fire protection efficiency of epoxy-based materials should be analyzed. Usually, the low temperature will destory the chemical structure of epoxy-based materials, resulting in the fire protection performance failure of epoxy-based materials. The author should be added the analysis of low temperature on the fire protection of epoxy-based materials. Moreover, the quantitative relationship between fire protection and cryogenic spillage protection for steel should be analyzed. moreover, the introduction should be enhanced.
Reviewer 2 Report
The paper is interesting and touch important aspects of fireproof insulation, but it is not everything clear for me. My comments are as follows:
- Acording to paper [23] the limit temperature is -200°C for cryiogenic and 1300°C for fire exposure. In the following part of the paper 500°C is taken as the limit temperature for the structure under fire condition. Please explain why? The detoriation of mechanical properties of steel is observed in 400°C
- In table 1 authors showed the characteristic of fireproofing coatings, in my opinion it should be fortified with thermal characteristic of insulation
- In the paper 3 types of profiles with different types of insulation were examined (Experiment 1,2,3). How many samples was in each single experiment? As I understood 2 specimens in experiment 1, and 1 in experiments 2 and 3. What is more all samples had different profiles and insulation, so how reliable are experiments? If it was more samples I think some statistical analysis should be included. Please add some comment
- Why profiles of specimens are different in each experiment? In such a case it is difficult to compare types of insulation.
- For each specimen authors give a profile ratio, but in the following part of paper there is any comment about that. Did authors find some relation between a profile ratio and some other results?
- For experiment 1 and 3 the limit criterion during cryiogenic fase was the temperature. In the case of experiment 2 the test was broken after 10 minutes. Please explain why?
- The thickness of coatings in experiment 1 is 18 or 27 mm. What is the technology of applying it, is it possible to get such accurate thickness?
- In the case of experiment 2 after fire exposure "the average temperature registered on the specimen was 468°C" (line 196). What was the highest noticed temperature, how big were the diffrences between all registered values.
- In the figure 4 temperature curves for specimens from experiment 1 have the shape that coresspond to fire curves shape. In the case of specimen 2 the character is completely different. What is your interpretation of this fact? What about specimen 3?
- In the conclusion authors indicated the need to develop methods that allow to assess the condition of fireproof coatings after cryogenic spillage impact. They indicated the need to aplly test according to ISO 20088-1:2016 and ISO 20088-3:2018 simultaneously, but in the paper different approach was applied. Why? Are authors going to continue experiments, if yes in what direction?
- Please improve conclusion taking into account my suggestions
Round 2
Reviewer 1 Report
The author carefully revised the manuscript, but the language should be improved.
